# Acculturation Gap Conflict and Self-Control Mediate the Association Between Negative Affect and Sleep Problems for Hispanic/Latino(a) Adolescents

**DOI:** 10.3390/ijerph22111722

**Published:** 2025-11-14

**Authors:** Manuel J. Bruzos, Benjelene D. Sutherland, Matthew T. Sutherland, Elisa M. Trucco

**Affiliations:** 1Center for Children and Families, Florida International University, Miami, FL 33199, USA; bsutherl@fiu.edu (B.D.S.); masuther@fiu.edu (M.T.S.); etrucco@fiu.edu (E.M.T.); 2Department of Psychology, Florida International University, Miami, FL 33199, USA; 3Department of Psychiatry, University of Michigan, Ann Arbor, MI 48109, USA

**Keywords:** negative affect, acculturation gap conflict, low self-control, sleep problems, adolescents

## Abstract

Background: Adolescent sleep problems are a public health concern, as roughly 60–70% of U.S. teens obtain less sleep than is recommended. Negative affect is a risk factor for adolescent sleep problems. However, factors explaining this association, to our knowledge, have not been explored in Hispanic/Latino(a) (H/L) adolescents. Acculturation gap conflict and low self-control may act as mediators linking negative affect to sleep problems. Methods: A sample of 223 H/L adolescents was assessed at three time points. We hypothesized that acculturation gap conflict and low self-control assessed at Wave 2 would mediate the association between negative affect at Wave 1 and sleep problems at Wave 3. Results: Negative affect predicted higher acculturation gap conflict, which in turn was associated with less self-control, and less self-control then predicted more sleep problems. Conclusions: These findings highlight acculturation gap conflict and low self-control as processes through which negative affect can impact later sleep problems in H/L adolescents. Elucidating modifiable factors impacting sleep health among H/L youth may inform intervention strategies. Targeting affect regulation, as well as improving parent-adolescent relationship quality within the context of acculturative differences, may help reduce sleep problems among this demographic group.

## 1. Introduction

Sleep problems are prevalent among teens. Approximately 60–70% of adolescents report not obtaining the recommended amount of sleep and experiencing sleep problems [1]. Such problems include having nightmares, sleeping less than most children their age, and feeling overtired [2]. This is concerning given that, if unaddressed, poor sleep could result in negative physical health consequences, including mental health problems and lower academic performance [3]. Later sleep timing has been associated with headaches, stomach aches, and back aches in adolescents [4]. Sleep problems may also negatively impact adolescents’ cognitive abilities, such as working memory and executive functioning [5,6], ultimately affecting academic performance [7]. As sleep health is essential for overall well-being [8], particularly among teens [9], elucidating factors impacting sleep may lead to strategies useful for augmenting mental and physical health.

Adolescents who identify as Hispanic/Latino (H/L) are an important point of focus because not only are they the fastest-growing minority population in the U.S. [10], but they are also more likely to experience sleep problems relative to non-H/L White youth [11,12]. This may partially relate to the fact that H/L adolescents experience greater cultural stress [13] and ethnic–racial discrimination [14]. More specifically, cultural stress has been found to be associated with less sleep duration and more sleep problems in H/L youth [15]. Ethnic–racial discrimination is also linked with a longer amount of time to fall asleep after lying down in bed, more sleep disturbance, daytime dysfunction, and next-day sleepiness [16]. As such, understanding mechanistic pathways contributing to sleep problems among H/L could help inform intervention strategies for this high-priority population.

Early temperamental differences can predict certain aspects of adolescent development and function. One temperamental aspect that has been linked with poor sleep is negative affect [17]. Negative affect is defined as feelings of emotional anguish, depression, anxiety, sadness, and other negative emotions [18]. Prior work demonstrates that negative affect experienced before sleep results in less slow-wave sleep, longer latency to slow-wave sleep onset, less rapid eye movement (REM) sleep, and more instances of awakenings [19]. Additionally, daytime emotional stress can influence sleep physiology, dream content, patterns, and nightmares [20]. Further, other correlates of negative affect, such as grief, loneliness, or hostility, are associated with increased sleep impairments [21]. While prior studies have considered interrelations between negative affect and sleep problems among adolescents [22,23], the mechanisms by which negative affect can influence sleep among H/L youth remain to be empirically examined. Acculturation gap conflict and low self-control are two plausible factors linking negative affect and sleep problems among this demographic group.

Negative affect can impact acculturative experiences (i.e., adapting to a new host culture) [24] and amplify child–parent conflicts [25]. Yet, the relation between negative affect and acculturation gap conflict (AGC) has received little empirical investigation. AGC, defined as discrepancies between the youths’ and parents’ level of acculturation, often displayed as the youth assimilating to a new host culture at a different pace than their parents, frequently leads to intergenerational conflicts [26,27,28]. More specifically, adolescents who are second-generation immigrants tend to acculturate more quickly than their first-generation parents [29]. Faster assimilation into a new culture can lead to bicultural stress characterized by day-to-day conflicts resulting from the pressure to adopt both the host and family’s native culture [30,31]. In turn, this assimilation gap may lead to conflict regarding differing cultural values between parents and children [32]. To the extent that AGC increases parent–child conflict, the interpersonal stress may lead to negative consequences, including poor sleep [26]. For example, acculturative stress [26] and parent–child conflict [33] are each independently associated with sleep problems. Acculturation stress has also been associated with inadequate sleep, poor sleep quality, and more difficulty falling asleep for H/L young adults and adults [34,35,36]. Further, family conflict that occurs between the ages of 7 and 15 has been shown to predict insomnia at age 18 [37]. As such, family conflict related to AGC is a plausible mechanism through which negative affect may impact sleep among H/L youth.

Low self-control is a second possible mediator linking negative affect and sleep problems. For example, low self-control among adolescents (i.e., deficits in the voluntary suppression of impulses) [38] is impacted by negative affect [39]. Negative emotions can also hinder executive functioning underlying self-control [40]. Although less work has focused on examining low self-control as a risk factor for sleep problems [41], prior work suggests it may be predictive of sleep problems [42,43]. Specifically, adolescents exhibiting high levels of impulsivity (a construct similar to low self-control) have greater difficulties maintaining a sleep routine relative to those low in impulsivity [41]. Further, youth with lower self-control are more likely to postpone their bedtime and, thus, less likely to go to bed at a planned time, often resulting in a suboptimal sleep duration [44]. To our knowledge, prior work has not characterized the interrelations between low self-control and sleep among H/L youth.

While research on the association between acculturation/acculturative stress and self-control in the H/L population has only appeared within the last decade, it is rather limited [45]. A notable exception pertains to work demonstrating that adolescents with higher acculturative stress also experienced lower levels of self-control [46]. Furthermore, AGC has been linked with certain problem behaviors characterized by low self-control (e.g., conduct problems) among H/L youth [26]. In turn, problem behaviors are known to have a negative impact on sleep problems [41]. That said, consideration of AGC, low self-control, and negative affect as etiological processes to sleep problems could have utility in informing preventative interventions targeting poor sleep health.

Given prior work, a reasonable hypothesis is that AGC and low self-control mediate the relation between negative affect and sleep problems among H/L adolescents. Moreover, AGC may be a more developmentally salient predictor than acculturative stress among adolescents, as parent–child conflict pertaining to cultural differences not only causes stress, but also predicts increased mental health problems [47]. The aim of this study was to longitudinally characterize the degree to which negative affect predicts subsequent sleep problems via AGC and low self-control in a serial mediation framework. We hypothesized that high levels of negative affect would be predictive of more AGC and that AGC would, in turn, be associated with lower levels of self-control, and finally that low self-control would predict more sleep problems.

## 2. Materials and Methods

### 2.1. Participants

The current study reflects a subsample of adolescents who participated in the Antecedents and Consequences of Electronic Nicotine Delivery Systems (ACE) Project, a longitudinal study examining e-cigarette use risk and protective factors within a community sample of high school freshmen and sophomores (*N* = 264). Study findings and additional participant and procedure details have been published previously [48,49,50]. Analyses focused on adolescents who identified as H/L and completed the Acculturation Gap Conflict Inventory (AGCI; *n* = 223; 53.4% girls, 91% White, Mage = 14.9). Those who identified as H/L and completed the AGCI measure (i.e., family identified with a culture other than American) did not differ from the larger study sample in terms of age. Yet, differences were found in terms of biological sex (*F*(1, 262) = 5.79, *p* = 0.02) and race (*F*(1, 262) = 9.14, *p* = 0.003), such that the subsample was characterized by more girls and White participants (W1 = 50.8% girls, 86.4% White vs. AGCI = 56% girls, 90.9% White). Those who completed W2 (*n* = 221) did not differ from those who completed W1 (*N* = 264) on any of the demographic characteristics. Lastly, differences were found in terms of biological sex among those who completed W3 (*n* = 133), such that more girls completed W3 (W2 = 53.4% girls vs. W3 = 60.9% girls; *F*(1, 219) = 8.62, *p* = 0.004). Data collection occurred from March 2018 through July 2022. Exclusionary criteria of the larger study consisted of adolescents with learning/intellectual disabilities, physical disabilities, neurological diseases, severe mental illness, and adolescent non-English fluency, as these may impact the ability for adolescents to provide appropriate assent, to understand and complete surveys, and/or the ability to tolerate study procedures.

### 2.2. Procedures

Local South Florida public schools were contacted about their willingness to participate in recruitment events. After the study personnel recruited participants through recruitment events at health fairs and schools, interested families completed an eligibility screen. Those meeting eligibility criteria were scheduled for an in-person, W1 data collection visit. After the consent/assent process, caregivers and adolescents completed questionnaires in separate testing rooms to enhance confidentiality. Questionnaires were administered on an iPad using REDCap [51,52]. Participants completed W2 assessments ~15 months after W1, and W3 assessments were ~12 months after W2. W2 and W3 consisted of similar procedures as W1; yet, due to COVID-19 safety restrictions, some participants completed questionnaires remotely on personal tablets or computers while assisted by research staff over Zoom to closely align with W1 procedures. Adolescent questionnaires took ~90 min to complete, while caregiver questionnaires took ~45 min. Caregivers and adolescents were both compensated for their participation. At each wave, caregivers received a gift card and cash reimbursement for travel, while adolescents received a separate gift card. The study was approved by Florida International University’s Institutional Review Board.

### 2.3. Measures

Early Adolescent Temperament Questionnaire-Revised (EATQ-R). To quantify negative affect at W1, the mean of three subscales was calculated as recommended by the measure developers [53]. Specifically, 7 items from the frustration subscale (e.g., “I get upset if I’m not able to do a task really well”), 6 items from the depressive mood subscale (e.g., “It often takes very little to make me feel like crying.”), and 6 items from the aggression subscale (e.g., “When I am angry, I throw or break things”) were averaged. Participants rated each item on a 5-point Likert scale (1 = Almost always untrue of you; 5 = Almost always true to you) with higher scores reflecting greater negative affect (Cronbach’s α = 0.83).

Acculturation Gap Conflict Inventory-Child (AGCI-C). AGC was quantified as the mean of 7 items (e.g., “My parents complain that I act too American”) from the adolescent-reported AGCI-C [26] administered at W2. Items were rated on a 7-point Likert scale (1 = Strongly disagree; 7 = Strongly agree) with higher scores reflecting greater conflict (Cronbach’s α = 0.86).

Youth Self Report (YSR). Raw scores from the YSR of the Achenbach System of Empirical Behavioral Assessment (ASEBA) [2] quantified low self-control at W2 and sleep problems at W3. Consistent with prior work [54,55,56], low self-control was assessed via the sum of 8 items (e.g., “I act without stopping to think”; Cronbach’s α = 0.73). Sleep problems were quantified by taking the sum of the following 5 items: “I have nightmares”, “I feel overtired without good reason”, “I sleep less than most kids”, “I sleep more than most kids during the day and/or night”, “I have trouble sleeping”. We note that the internal consistency of the sleep problems measure was low (Cronbach’s α = 0.51), consistent with prior work utilizing this scale among adolescents and children [57,58,59,60]. Items on the YSR were rated on a 3-point Likert scale (0 = Not true, 2 = Very true or often true). The YSR was administered during all three waves, allowing for low self-control and sleep problems at W1 to be utilized as covariates. Youth age and biological sex were also used as covariates.

### 2.4. Data Analytic Plan

We calculated descriptive statistics and correlations for all study variables. Structural equation modeling was performed in R Studio version 2025.5.1.513 [61] to estimate serial mediation with two mediators employing the “sem” package [62]. Specifically, the model tested whether adolescents’ self-reported AGC (M1) and self-control at W2 (M2) mediated the association between negative affect at W1 (X) and sleep problems at W3 (Y). All study variables were normally distributed (skewness range = 0.05–1.26, kurtosis range = −0.53–2.06). Indirect effects were estimated using bootstrap confidence intervals (CIs) as a rigorous method for estimating mediated effects [63]. Full information maximum likelihood was also estimated to address missing data.

## 3. Results

Table 1 provides the means, standard deviations, and correlations for all study variables. Consistent with prior work [19,24,25,39], negative affect at W1 was positively correlated with AGC at W2, low self-control at both W1 and W2, and sleep problems at both W1 and W3. AGC was positively correlated with low self-control at W2 but not with sleep problems at W3. Low self-control at W2 was positively correlated with sleep problems at W3. Notably, girls were more likely to report low self-control at W1, negative affect at W1, and sleep problems at W3. Boys were more likely to report high AGC at W2.

### Serial Mediation Model Estimating the Impact of Negative Affect on Sleep Problems

We observed a significant indirect effect supporting serial mediation when considering the influence of negative affect on future sleep problems via AGC and low self-control (Table 2, Figure 1). The model accounted for ~13% of the variance in AGC. Negative affect at W1 was a significant predictor of AGC at W2 (effect = 0.51, *p* = 0.002), such that higher negative affect predicted more AGC. Further, the model accounted for ~45% of the variance in low self-control. AGC at W2 was associated with low self-control at W2 (effect = 0.43, *p* = 0.02), such that more conflict was linked with lower self-control. Lastly, the model accounted for ~35% of the variance in sleep problems. Low self-control at W2 predicted sleep problems at W3 (effect = 0.05, *p* = 0.000), such that lower self-control predicted more sleep problems. Notably, the direct effect between negative affect and sleep problems was not significant (effect = 0.02, *p* = 0.77). Moreover, the simple mediated effects were not supported in this model. Taken together, the model supported an indirect effect of negative affect on sleep problems through AGC and self-control (indirect effect = 0.01, CI [0.001, 0.023]). 

Given that acculturation gap conflict and low self-control were assessed at the same time point (i.e., W2), post hoc analyses were conducted to elucidate possible bidirectional associations between these variables. We examined if negative affect at W1 predicted low self-control at W2 (effect = 0.57, *p* = 0.17), and if, in turn, low self-control was associated with acculturation gap conflict at W2 (effect = 0.05, *p* = 0.07). Lastly, we tested whether acculturation gap conflict predicted sleep problems at W3 (effect = 0.01; *p* = 0.74). Findings did not support a significant serial mediation indirect effect (indirect effect = 0.00, CI [−0.002, 0.005]).

## 4. Discussion

Proper sleep is critical for supporting healthy physical, emotional, and cognitive development in adolescents [64,65]. However, most adolescents experience sleep problems [1]. Notably, H/L adolescents tend to experience more sleep problems (i.e., insomnia) compared to non-H/L White adolescents [11,12]. As such, the current study elucidated mechanistic pathways to sleep problems among H/L adolescent youth. Consistent with our hypothesis, AGC and low self-control mediated the association between negative affect and sleep problems. Specifically, greater negative affect led to more AGC, which in turn was associated with less self-control. Less self-control, then, predicted future sleep problems. While we observed a significant positive correlation between negative affect and future sleep problems (Table 1), a direct effect linking these constructs was not detected in the serial mediation model. As prior work indicates negative affect can impact future sleep problems [19,66], our findings extend the current literature by highlighting AGC and self-control as mechanistic processes underlying this association among H/L youth. Interventions targeting AGC and self-control could offset risks for later sleep problems that disproportionately impact this demographic group.

AGC research is scant, particularly among H/L adolescents [67,68]. This is suboptimal because AGC can negatively impact H/L adolescents, families, and communities. Our results indicate that negative affect increases AGC. This link is likely enhanced during adolescence, a developmental period notable for reorganization in affective systems given myriad cognitive, neurobiological, and social changes [69], as well as enhanced potential for parent–child conflicts as adolescents establish greater autonomy, greater independence, and grapple with emerging identities [70]. Prior work indicates that negative affect regulation plays an important role in interpersonal conflict. Effective familial conflict management often requires the integration of contextual information from various sources, regulation of emotions, and deployment of various coping skills [71]. It follows that H/L adolescents who are high in negative affect are more likely to experience acculturation gap conflict, given the adverse impact of negative affect on parent–child relationships generally and to their conflict interactions in particular. More specifically, negative affect is associated with increased perceived stress [72,73,74,75]. These symptoms of perceived stress could negatively impact the parent–child relationship [25] and increase AGC. For example, high negative affect may contribute to adolescents feeling more uncomfortable about choosing one culture over another [26].

Consistent with prior work [26], AGC was linked with low self-control. Although temperamental traits are typically viewed as biologically based and stable [76], self-control can be more malleable during periods of change in an individual’s biology and social ecology, such as adolescence. During adolescence, self-control develops through an interplay between early childhood temperament and socialization [77]. Socialization, especially within the family context, is largely a cultural process [78]. Specifically, socialization within a strong collectivistic orientation (e.g., traditional H/L cultures) is characterized by the values of “familismo” and “respeto” [79]. As such, elevated parent–child AGC could lead to family distancing [70], which could disproportionately impact the development of effective self-control among H/L adolescents. These life experiences could also bring about physiological responses due to stress, negatively affecting prefrontal cognitive function, which could alter motivational pathways that affect one’s impulse control [80].

Our findings also demonstrate that low self-control impacts sleep problems assessed a year later among H/L youth. This is consistent with prior work indicating that low self-control and related constructs (e.g., impulsivity) can increase sleep problems [41,42,44]. Low self-control is characterized by impulsive actions and seeking instant gratification over long-term goals [38]. For example, prioritizing other activities (e.g., watching one more episode of their favorite show) over bedtime in the short-term can have a negative impact on sleep at night and next-day functioning [44]. Interestingly, it has been demonstrated that individuals are more likely to engage in bedtime procrastination, defined as delaying going to bed in order to engage in activities that provide hedonic reward, despite causing issues later on [81], after subjectively more stressful days taxing self-regulatory resources [82]. Ironically, adolescents would benefit the most from getting a good night’s sleep on more challenging days, which further impairs their ability to regulate emotions and navigate stressors the next day [83]. Over time, this vicious cycle could contribute in part to the disparities in poor sleep health among H/L adolescents.

Our findings enhance the etiological understanding of factors impacting sleep among H/L adolescents, which informs both transcultural and culture-specific clinical practices. Transculturally, intervention efforts focused on negative affect reduction (e.g., brief mindfulness training) may reduce subsequent AGCs with parents. Mindfulness training is characterized by training an individual to be more presently aware of their current experiences by activating attention and emotion regulation skills [84]. Recent meta-analytic work shows that brief mindfulness training can have an immediate and substantive impact on decreasing negative affectivity in both clinical and nonclinical samples [85]. Culturally, intervention efforts targeting acculturative stress and/or parent–child conflict for H/L adolescents, similar to Entre Dos Mundos [86], could help mitigate AGC, and subsequent low self-control leading to sleep problems. Specifically, Entre Dos Mundos brings together groups of adolescents and families to discuss stressors and challenges relating to acculturation using Bicultural Effectiveness Training (BET). Reducing conflict among caregivers and their adolescents can improve family dynamics, as well as enhance an adolescent’s functioning and well-being [87]. Promoting self-control may also have utility in improving sleep hygiene, such as putting restrictions on technology use before bedtime and implementing bedtime routines [88]. Psychoeducation on effective sleep hygiene practices and promoting sleep routines may be particularly important for H/L families. For example, one report demonstrated that 74% of H/L children, as compared to 22% of White children, had a television in their bedrooms, with a commonly cited reason for their presence being to help the child fall asleep [89]. Taken together, our findings identify multiple modifiable targets that, if addressed, could result in breaking the pernicious loop that contributes to poor sleep health disparities among H/L adolescents.

The current study provides a deeper understanding of unique etiological factors contributing to poor sleep health among H/L adolescents. Nevertheless, limitations should be noted. First, while our sample likely represents typical high school students within the study catchment area, the sample reported generally high levels of impulsivity. The larger study required adolescents to meet criteria for personality factors linked with substance use, such as high impulsivity and sensation seeking. Yet, only a small portion of the sample (2.6%) did not meet eligibility criteria, which suggests that our sample is likely representative. We also controlled for low self-control at baseline in our mediation model. Second, excluding adolescents with non-English fluency could have limited representation of less acculturated youth potentially biasing findings related to acculturation gap conflict. Third, AGC and low self-control were assessed at the same time point; bidirectional associations likely exist between these variables. Although post hoc analyses suggested that low self-control did not influence AGC, future work should consider an additional time point to account for temporal precedence. Fourth, our study used self-report measures to assess negative affect, AGC, low self-control, and sleep problems. Adolescents may have biased perceptions of their emotional states. Prior work recommends assessing adolescent self-control and negative affect across multiple reporters as self-reports may be especially biased among adolescents low in self-control and high in negative affect [90,91]. Yet previous work indicates that caregiver reports of adolescent self-control and negative affect could also be biased, given parents’ own self-control and mental health difficulties, and that the accuracy of caregiver reports may decrease as youth transition to adolescence, given less time spent in the home [91,92]. The use of objective measures, such as actigraphy and polysomnography to collect sleep data, may also be useful in reducing bias or misperceptions. Yet perceptions are important to measure as well, given that they could provide researchers with unique information, such as perceived sleep quality, that may not be captured by objective measures. Furthermore, shared method variance may boost the association across these constructs. Therefore, incorporating collateral reports and more objective measures of these constructs may reduce bias and shared method variance. Fifth, there are likely important biological sex differences that could have affected the results. For example, consistent with our findings, negative affect tends to be higher for girls than boys [93]. As the sample size in the current study was too small to formally test biological sex differences through more complex mediation models (i.e., moderated mediation), it will be important for future studies to determine whether these mechanistic processes operate similarly for boys and girls. Sixth, we were not able to control for a number of important factors that have been associated with sleep quality and quantity (e.g., socioeconomic status [94], body mass index [95], puberty [96], physical activity [97], exposure to light [98], and later chronotype [99]). Lastly, we examine sleep problems more generally, whereas later chronotype and low circadian system robustness also encapsulate poor sleep health constructs that are linked to worse health and well-being [100,101]. This is especially true during the adolescent years, whereby the typical chronotype shift towards later sleep and wake times has been linked to enhanced risk-taking and poor mental health [102]. These factors were not included in the present study and should be considered by future research.

## 5. Conclusions

As adolescent sleep problems are considerably high, particularly for H/L adolescents, the current study provides detailed insight into what factors are linked to H/L adolescents’ sleep problems. More specifically, the present study found a significant association between negative affect and adolescent sleep problems through acculturation gap conflict and low self-control. H/L adolescents who endorse high levels of negative affect experienced more acculturation gap conflict, which in turn led to lower self-control, and low self-control was predictive of future sleep problems. Interventions, both transcultural and culture-specific, targeting affective regulation strategies, acculturative stress and parent–child conflict, self-control, and sleep hygiene can help reduce sleep problems for H/L adolescents. Preventive interventions that focus on addressing these modifiable targets could ultimately offset disparities in sleep health reported among H/L youth.

## Figures and Tables

**Figure 1 ijerph-22-01722-f001:**
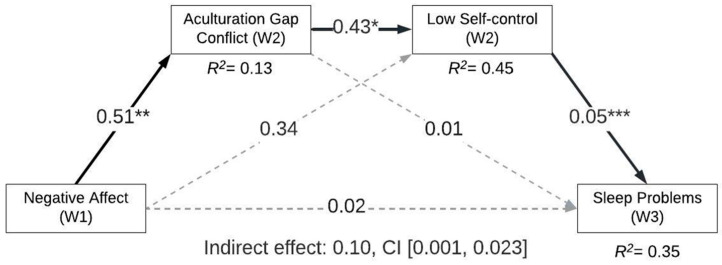
Serial mediation model linking negative affect with sleep problems via acculturation gap conflict and low self-control. Note: * *p* < 0.05; ** *p* < 0.01; *** *p* < 0.001; W1 = Wave 1; W2 = Wave 2; W3 = Wave 3; estimated covariates (not depicted): Age W1, biological sex W1, low self-control W1, and sleep problems (W1). Estimated coefficients are unstandardized.

**Table 1 ijerph-22-01722-t001:** Means, standard deviations, and correlations for study variables.

	M	SD	Correlations							
Study Variables			1	2	3	4	5	6	7	8
Age (W1)	14.92	0.68	−							
2. Biological Sex ^a^ (W1)	0.47	0.50	0.04	−						
3. Low Self-Control (W1)	5.54	3.03	−0.01	**−0.14 ***	−					
4. Sleep Problems (W1)	0.58	0.39	0.01	−0.08	**0.52 *****	−				
5. Negative Affect (W1)	2.59	0.52	0.08	**−0.17 ***	**0.57 *****	**0.48 *****	−			
6. Acculturation Gap Conflict (W2)	2.16	1.13	−0.02	**0.27 *****	0.07	0.00	**0.19 ***	−		
7. Low Self-Control (W2)	5.11	3.10	−0.05	−0.02	**0.62 *****	**0.45 *****	**0.39 *****	**0.21 ****	−	
8. Sleep Problems (W3)	0.53	0.39	0.03	**−0.24 ***	**0.41 *****	**0.46 *****	**0.37 *****	0.08	**0.51 *****	−

Note: Correlation coefficients in bold represent significant associations. * *p* < 0.05; ** *p* < 0.01; *** *p* < 0.001; W1 = Wave 1; W2 = Wave 2; W3 = Wave 3; ^a^ Girl = 0, Boy = 1.

**Table 2 ijerph-22-01722-t002:** Regression coefficients, standard errors, and summary information of the negative affect serial mediation model predicting sleep problems.

	Acculturation Gap Conflict W2 (M_1_)				Low Self-Control W2 (M_2_)				Sleep Problems W3 (Y)			
	Coefficient	(95% CI)	*SE*	*z* Value	Coefficient	(95% CI)	*SE*	*z* Value	Coefficient	(95% CI)	*SE*	*z* Value
Intercept	1.25	(−2.67, 5.31)	2.10	0.59	4.23	(−4.03, 12.58)	4.08	1.04	−0.35	(−1.82, 1.05)	0.71	−0.49
Age	−0.05	(−0.32, 0.22)	0.14	−0.34	−0.29	(−0.84, 0.25)	0.27	−1.07	0.03	(−0.07, 0.13)	0.05	0.63
Biological Sex ^a^	**0.70 *****	**(0.32, 1.04)**	**0.18**	**3.82**	0.14	(−0.68, 0.93)	0.41	0.33	−0.120	(−0.240, 0.006)	0.07	−1.87
Low Self-Control (W1)	-	-	-	-	**0.64 *****	**(0.49, 0.77)**	**0.08**	**8.34**	-	-	-	-
Sleep Problems (W1)	-	-	-	-	-	-	-	-	**0.27 ***	**(0.06, 0.49)**	**0.11**	**2.39**
Negative Affect (W1)	**0.51 ****	**(0.20, 0.86)**	**0.16**	**3.10**	0.34	(−0.46, 1.17)	0.41	0.82	0.02	(−0.13, 0.17)	0.07	0.30
Acculturation Gap Conflict (W2)	-	-	-	-	**0.43 ***	**(0.03, 0.76)**	**0.18**	**2.42**	0.01	(−0.07, 0.08)	0.03	0.37
Low Self-Control (W2)	-	-	-	-	-	-	-	-	**0.05 *****	**(0.02, 0.07)**	**0.01**	**4.05**
	***R^2^* = 0.13**				***R^2^* = 0.45**				***R^2^* = 0.35**			

Note: Regression coefficients in bold represent significant associations. * *p* <0.05, ** *p* <0.01, *** *p* <0.001; W1 = Wave 1, W2 = Wave 2, W3 = Wave 3; ^a^ Girl = 0, Boy = 1. Estimated coefficients are unstandardized.

## Data Availability

Research protocols, deidentified and processed data, code, and study materials will be made available upon request.

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
