# Peer review of "Acculturation Gap Conflict and Self-Control Mediate the Association Between Negative Affect and Sleep Problems for Hispanic/Latino(a) Adolescents"

_ijerph, 2025, doi:10.3390/ijerph22111722_

Round 1
Reviewer 1 Report
Comments and Suggestions for Authors
Thank you for the opportunity to review this manuscript. This study addresses an important and timely topic by examining psychosocial processes that contribute to sleep health among Hispanic/Latino(a) adolescents, a population often underrepresented in developmental and health research. Its focus on acculturation gap conflict and self-control as mediating mechanisms offers valuable insight into culturally specific pathways linking emotional well-being and sleep outcomes.
Introduction
The introduction is well written, well documented and provides the reader with relevant information of what is known and the research gaps, proving the significance of this paper and how it contributes to the knowledge base.
Methods
In the text, I would recommend to always refer to “sex” as “biological sex”, if pertinent.
Has the “longitudinal study examining e-cigarette use risk and protective factors within a community sample of high school freshmen and sophomores 129 (N=264) that consisted predominantly of H/L adolescents” been published? If so, please include the citation of that published paper and include a few words emphasizing that the work has been published.
Was compensation provided to caregivers? How was it provided, using e-cards? Please clarify.
The corresponding IRB was that of FIU or UM? Please clarify.
Including only English-speaking adolescents may have limited representation of less acculturated youth and potentially biased findings related to acculturation gap conflict. Could the authors clarify their rationale for this criterion?
Authors mention “recruitment events”. Were these events at the schools or where did these events take place?
No comments on the Results’ section
Discussion
I identified a couple of sentences that lack supporting references and would benefit from being backed by empirical evidence and clarified for completeness:
- “Sleep is vital for development.” This is the opening sentence for the discussion, while this is widely accepted, clarifying and adding a reference would make it better. For instance, something like: Proper sleep is crucial for supporting physical, emotional, and cognitive development in adolescents, and adding a reference.
- “AGC research is scant particularly among H/L adolescents”. Add supporting reference.
Reviewer 2 Report
Comments and Suggestions for Authors
This paper outlines a compelling and significant longitudinal study addressing an underexplored area in Hispanic/Latino (H/L) adolescent sleep health. The research offers considerable strengths, primarily its focus on modifiable mediators—acculturation gap conflict and low self-control—which are vital for informing targeted interventions. The clear hypothesis-driven approach, supported by three-wave data, provides a strong foundation for the proposed mediation pathways. The identified practical implications for affect regulation and parent-adolescent relationships are both relevant and actionable for this demographic. Overall, this is a valuable contribution can be recommended. However, there are several areas that warrant further consideration and discussion:
-
The sample size of 223, while acceptable limits statistical power for more complex mediation models. This should be included.
-
The reliance on self-reported measures inherently introduces the potential for bias, with speciific significant concern on the potential for confounding variables, such as socioeconomic factors, that were not explicitly addressed.
-
More critically, given the study’s focus on sleep, the absence of key factors that directly impact sleep quality and quantity is a notable limitation. Related core factors include light exposure (especially artificial light sources), physical activity, and chronotype.
-
Notably, later chronotypes in association with compromised circadian health and sleep were repeatedly noticed in Spain and Portugal (e.g. doi: 10.5935/1984-0063.20180036; doi: 10.3390/biology11081130), Uruguay, and Argentina (e.g. https://doi.org/10.1016/j.neuroscience.2025.02.022). Since these were not included as co-factors in the present work, their influence on the observed relationships must be thoroughly discussed.
